# S-acylation of P2K1 mediates extracellular ATP-induced immune signaling in *Arabidopsis*

Dongqin Chen [1✉], Fengsheng Hao[1], Huiqi Mu[1], Nagib Ahsan[2,4], Jay J. Thelen [2] & Gary Stacey [2,3✉]

S-acylation is a reversible protein post-translational modification mediated by protein S-acyltransferases (PATs). How S-acylation regulates plant innate immunity is our main concern. Here, we show that the plant immune receptor P2K1 (DORN1, LecRK-I.9; extracellular ATP receptor) directly interacts with and phosphorylates *Arabidopsis* PAT5 and PAT9 to stimulate their S-acyltransferase activity. This leads, in a time-dependent manner, to greater S-acylation of P2K1, which dampens the immune response. *pat5* and *pat9* mutants have an elevated extracellular ATP-induced immune response, limited bacterial invasion, increased phosphorylation and decreased degradation of P2K1 during immune signaling. Mutation of S-acylated cysteine residues in P2K1 results in a similar phenotype. Our study reveals that S-acylation effects the temporal dynamics of P2K1 receptor activity, through autophosphorylation and protein degradation, suggesting an important role for this modification in regulating the ability of plants in respond to external stimuli.

[1] State Key Laboratory of Agrobiotechnology, College of Plant Protection, China Agricultural University, Beijing, China. [2] Division of Biochemistry, C.S. Bond Life Science Center, University of Missouri, Columbia, MO, USA. [3] Divisions of Plant Science, C.S. Bond Life Science Center, University of Missouri, Columbia, MO, USA. [4]Present address: Department of Chemistry and Biochemistry, The University of Oklahoma, Norman, OK, USA. ✉email: chendq@cau.edu.cn; staceyg@missouri.edu

Plants are sessile organisms that rely exclusively on innate immunity to defend against detrimental microbes or pests, unlike mobile animals that possess adaptive immune systems. In order to perceive and trigger innate immune responses, plants employ a two-tier innate immune system, which includes critical plasma membrane (PM) localized receptors[1]. The first layer of defense is termed pattern-triggered immunity (PTI), since it relies on PM, pattern-recognition receptors (PRRs), which recognize conserved, pathogen-associated molecular patterns (PAMPs). Pathogens can also cause cell damage releasing damage-associated molecular patterns (DAMPs) that are also recognized by PM-localized, PRRs[2]. PTI can be defeated by pathogen-produced, effector proteins that target a variety of cellular components involved in innate immunity. Hence, the second layer of defense, termed effector trigger immunity (ETI), involves plant recognition systems that counter the action of these pathogen effector proteins[3].

PRRs encompass a variety of receptor-like kinase (RLK) subfamilies. Receptor-like kinases comprise a ligand-binding ectodomain, a transmembrane domain and an intracellular kinase domain. For example, FLS2 (FLAGELLIN SENSITIVE 2), containing a leucine rich repeat (LRR) ectodomain, recognizes bacterial flagellin via direct binding to a conserved 22-aminoacid epitope, flg22[4–6]. Another well-studied plant LRR-RLK is the Arabidopsis EFR receptor, which recognizes EF-Tu via perception of the conserved N-acetylated epitope elf18[7,8]. A second family of RLKs involved in PAMP recognition is the lysin motif (LysM-RLK) family, which includes CERK1 (CHITIN ELICITOR RECEPTOR KINASE 1). Chitin is a major constituent of fungal cell walls, which is directly bound by AtCERK1 (AtLYK1) and AtLYK5 in Arabidopsis[9,10]. Two plant LRR-RLK members, PEPR1 and PEPR2, were shown to specifically recognize AtPEP peptides released during cell/tissue damage[11,12]. Extracellular ATP (eATP) is also released during tissue damage or in response to specific elicitation, including pathogens[13]. A member of the lectin-RLK subfamily, P2K1 (DORN1), was shown to be the key receptor that recognizes eATP resulting in the induction of an innate immunity response[14]. P2K1 (LecRK-I.9) was identified as a positive regulator of plant defense against the oomycete pathogens, Phytophthora brassicae, Phytophthora infestans, and bacterial pathogen Pseudomonas syringae DC3000[15–18]. On the other hand, eATP can elicit P2K1-mediated RBOHD phosphorylation to regulate stomatal aperture with important implications for regulating plant photosynthesis, water homeostasis, pathogen resistance, and ultimately yield[19].

While different PRRs recognize different PAMPs/DAMPs, the plant responses induced upon activation of these receptors are highly similar. For example, PTI is characterized by an elevation of cytoplasmic calcium ($Ca^{2+}$), reactive oxygen species (ROS), nitric oxide (NO), the activation of mitogen-activated protein kinases (MAPKs), and the expression of immune related genes[13,20]. Activation of the PRR by its cognate ligand leads to autophosphorylation and transphosphorylation of other proteins that is often followed by receptor ubiquitination and degradation, which subsequently dampens the immune response[21].

In contrast to phosphorylation, other forms of protein covalent modification are poorly studied in plants, although they are known to occur. For example, S-acylation, a reversible acylation of cysteine residues via a thioester linkage, is well characterized in mammals but poorly studied in plants. Mammalian studies have shown a critical role for S-acylation in controlling PM association, subcellular trafficking, stability, protein-protein interactions, enzymatic activity, and many other functions[22–24]. Dynamic protein S-acylation is catalyzed by protein S-acyl transferases (PATs) that contain a conserved Asp-His-His-Cys (DHHC) catalytic domain, while deacylation is mediated by acyl protein thioesterases (APTs)[24]. In human, a family of 24 DHHC-PATs was identified and shown to be involved in many physiological processes, as well as diseases spanning from neuropsychiatric disorders to cancers, cellular differentiation, melanomagenesis, and so on[24–26]. Most human DHHC-PATs are localized to endomembrane compartments such as Golgi, endosomes, endoplasmic reticulum, with only two proteins, DHHC20 and DHHC21, found at that the PM, which mediate epidermal growth factor receptor (EGFR) signaling in cancer and inflammatory responses[27,28].

The genome of the genetic model plant, Arabidopsis thaliana, encodes 24 DHHC-PATs[29]. However, in contrast to humans, 12 Arabidopsis DHHC-PATs are localized to the PM, perhaps underlying the critical role that PM RLKs play in the ability of plants to respond to their environment[24,29]. There are relatively few studies describing the function of PATs in plants. Of the 24 encoded AtPATs, only seven have been studied and shown to play a role in root hair growth[30,31], cell death, ROS production[32,33], salt tolerance[34], cell expansion and division[35], male and female gametogenesis[36], branching[37], and early seedling growth and establishment[38]. Although a few plant proteins were previously shown to be S-acylated, including FLS2[39], specific substrates for these AtPATs are rare, while their biological functions or mechanisms are also unknown.

In this manuscript, we provide evidence for a model in which the dynamics of autophosphorylation, S-acylation, and receptor turnover control the ability of the P2K1 receptor to elicit an immune response. In the presence of eATP, P2K1 rapidly autophosphorylates, with a concomitant accumulation in S-acylation, and, ultimately, turnover of the receptor from the PM. Ligand activation of P2K1 leads to phosphorylation of PAT5 and PAT9 that restores S-acylation of the P2K1 receptor, re-establishing the inactive, steady-state form of the receptor.

## Results

**PAT5 and PAT9 restrict the plant immune response.** In our previous study, we used a mass spectrometry-based in vitro phosphorylation strategy, termed kinase client assay (KiC assay)[40], to identify putative substrates of the P2K1 kinase domain; such as the NADPH oxidase respiratory burst oxidase homolog D (RBOHD)[19]. In addition, this assay also identified two homologous genes, AtPAT5 and AtPAT9, which encode DHHC-PATs proteins likely involved in protein S-acylation (Fig. 1a). Both PAT5 and PAT9 proteins were localized to the PM (Supplementary Fig. 1a, b), consistent with possible interaction with PM-localized P2K1.

To determine whether PAT5 and PAT9 are involved in immune signaling by the P2K1 receptor, we investigated the phenotypes of pat5 or pat9 mutants by examining the response to ATP. Interestingly, the rapid elevation of cytoplasmic $Ca^{2+}$ and ROS burst after ATP treatment were significantly increased in both mutants compared to the wild type (Col-0; Fig. 1b, c and Supplementary Fig. 1e). Given their possible roles in modulating plant receptor activity, we also tested these immune responses with other elicitors, including flg22 and chitin. Similar results were observed using flg22 or chitin as the elicitor (Supplementary Fig. 1c, d, f, g), suggesting that PAT5 and PAT9 might impact other PRRs. Consistent with the increase in cytoplasmic $Ca^{2+}$ and ROS, ATP, flg22, or chitin-triggered MAPKs activation was also higher in the pat5 and pat9 mutants relative to the Col-0 wild type, particularly at 30 and 60 min when wild-type MAPK activation was decreasing (Fig. 1d and Supplementary Fig. 1i, j). Consistent with these stronger immune responses, growth of the bacterial pathogen Pseudomonas syringae DC3000 (Pst. DC3000) after surface inoculation was significantly reduced in the pat5 or

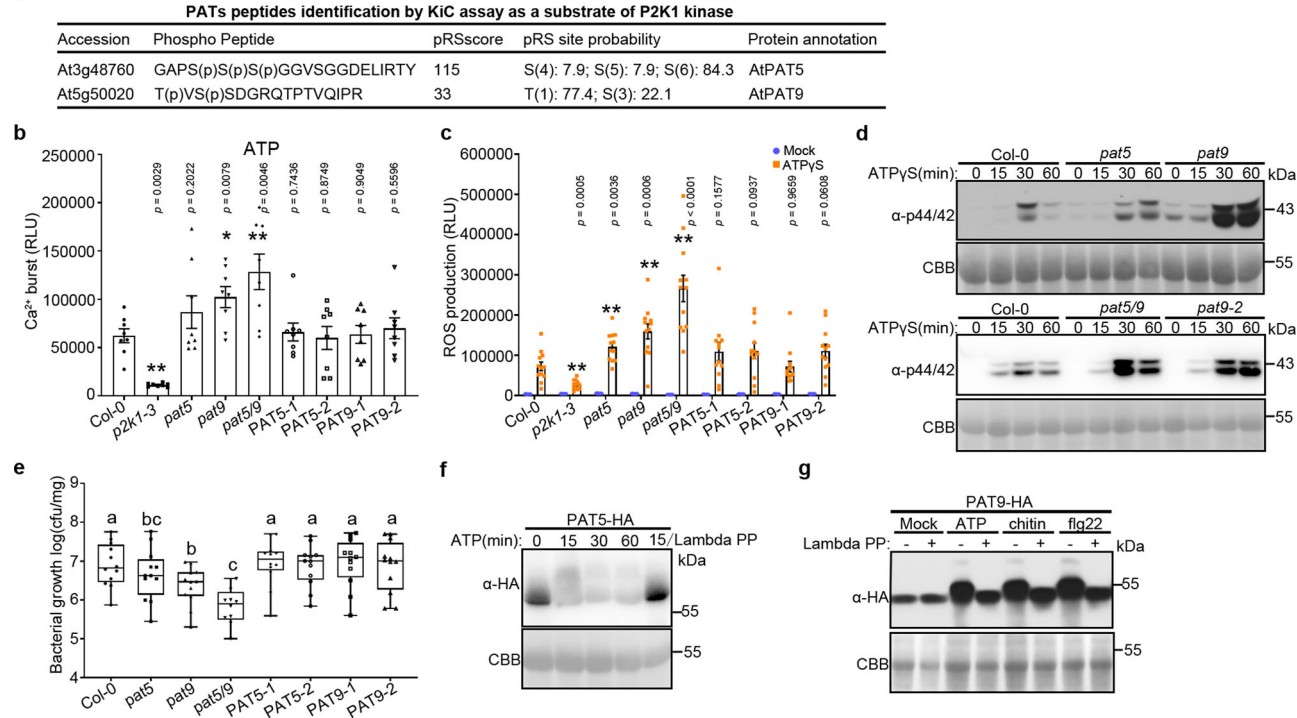

**Fig. 1 PAT5 and PAT9 control PTI response triggered by eATP. a** Identification of PAT5 and PAT9 tryptic peptides as a substrate of P2K1 kinase by KiC assay. **b** Ligand-induced calcium influx. 5-day-old seedlings were treated with 100 µM ATP for 15 min. RLU, relative luminescence units. Error bars indicate ±SEM; $n = 8$ seedlings; *$p < 0.05$, **$p < 0.01$, P-values indicate significance relative to Col-0 and were determined by one-sided ANOVA with unpaired, two-tailed Student's $t$ test. **c** ROS production was measured in leaf disks after treatment with 200 µM ATPγS for 30 min. Leaf disks were taken from WT (Col-0), pat5, pat9, and pat5/9 double mutants, or their complemented lines PAT5 (NP::ATPAT5-HA/Atpat5) and PAT9 (NP::ATPAT9-HA/Atpat9). Error bars indicate ±SEM; $n = 12$ leaf disks; *$p < 0.05$, **$p < 0.01$, P-values indicate significance relative to Col-0 with ATPγS treatment and were determined by one-sided ANOVA with unpaired, two-tailed Student's $t$ test. **d** MAPKs activation of Arabidopsis leaf disks that treated with 200 µM ATPγS for the times indicated. Coomassie Brilliant Blue (CBB) staining of protein was used as loading control. **e** PAT5 and PAT9 negatively mediated bacterial invasion. Arabidopsis seedlings with the indicated genotype ($x$ axis) were flood-inoculated with Pst. DC3000 bacteria and bacterial growth determined by plate counting 3 days post inoculation. Error bars indicate ±SEM; $n = 12$ (biological replicates); means with different letters are significantly different ($p < 0.05$); P-values indicate significance relative to Col-0 and were determined by one-sided ANOVA with multiple comparisons and adjusted using Benjamini–Hochberg post-test. Box extends from the 25th to the 75th percentile, whiskers denote minima and maxima (Boxplots, Col-0: max = 7.76; min = 5.88; center = 6.82; Q2 (25%) = 6.45; Q3 (75%) = 7.43, pat5: max = 7.76; min = 5.45; center = 6.62; Q2 (25%) = 6.13; Q3 (75%) = 7.05, pat9: max = 6.97; min = 5.3; center = 6.47; Q2 (25%) = 6.1; Q3 (75%) = 6.71, pat5/9: max = 6.55; min = 5; center = 5.9; Q2 (25%) = 5.49; Q3 (75%) = 6.21, PAT5-1: max = 7.7; min = 5.59; center = 7.05; Q2 (25%) = 6.76; Q3 (75%) = 7.25, PAT5-2: max = 7.64; min = 5.85; center = 7; Q2 (25%) = 6.51; Q3 (75%) = 7.15, PAT9-1: max = 7.72; min = 5.6; center = 7.09; Q2 (25%) = 6.58; Q3 (75%) = 7.47, PAT9-2: max = 7.7; min = 5.78; center = 7; Q2 (25%) = 6.27; and Q3 (75%) = 7.47). Experiments were repeated three times with similar results. **f, g** Ligand triggers PAT5 and PAT9 phosphorylation after treated with 200 µM ATP, 1 µM flg22, or 50 µg/ml chitin in their complemented lines PAT5 and PAT9. Lambda protein phosphatase (Lambda PP, − and +) was added to release phosphate groups. CBB was used as loading control. All experiments were repeated and analyzed three times with similar results.

pat9 mutant plants (Fig. 1e and Supplementary Fig. 1h). These phenotypes were increased significantly in pat5/9 double mutant plants (Fig. 1b, c, e, and Supplementary Fig. 1c, d, f, g), indicating that the functions of PAT5 and PAT9 are partially redundant.

In order to confirm PAT5 and PAT9 mediated PTI responses, we expressed the full-length PAT5 or PAT9 proteins, driven by their native promoters, in pat5 and pat9 mutants (Supplementary Fig. 1k), respectively. Expression of these proteins fully complemented the pat5 and pat9 mutant phenotype, restoring a wild-type PTI response (Fig. 1b, c, e and Supplementary Fig. 1 c, d, f, g). Furthermore, we tested whether the activation of the PAT5 and PAT9 proteins was associated with these PTI responses. Indeed, addition of ATP, flg22, or chitin-triggered phosphorylation of PAT5 and PAT9 (Fig. 1f, g and Supplementary Fig. 1l). Taken together, the above findings reveal that PAT5 and PAT9 are functionally redundant and act to negatively regulate plant innate immunity mediated by PAMP/DAMP, including ATP, flg22, and chitin.

The Arabidopsis PAT proteins are mainly clustered in three groups[29], while PAT5, PAT6, PAT7, PAT8, and PAT9 are in the same clade and show high sequence conservation (Supplementary Fig. 2a). We next examined their gene expression upon eATP and pathogen treatment. Quantitative RT-PCR (qRT-PCR) analysis showed that PAT5, PAT6, PAT9, and P2K1 were significantly upregulated upon ATP treatment after 30 min or 1 h compared with 0 h in wild type (Col-0; Supplementary Fig. 2b). Meanwhile, the expression of PAT5 and P2K1 was also greatly increased 1 or 2 h after inoculation with Pst DC3000 (Supplementary Fig. 2b). These data indicate that eATP or pathogen can induce PAT5 to PAT9 clade gene expression, suggesting they may be functionally redundant.

**PAT5 and PAT9 directly interact with P2K1.** To verify the KiC assay result that PAT5 and PAT9 are the phosphorylation substrates of the P2K1 receptor (ATP-elicited PTI), we used a variety

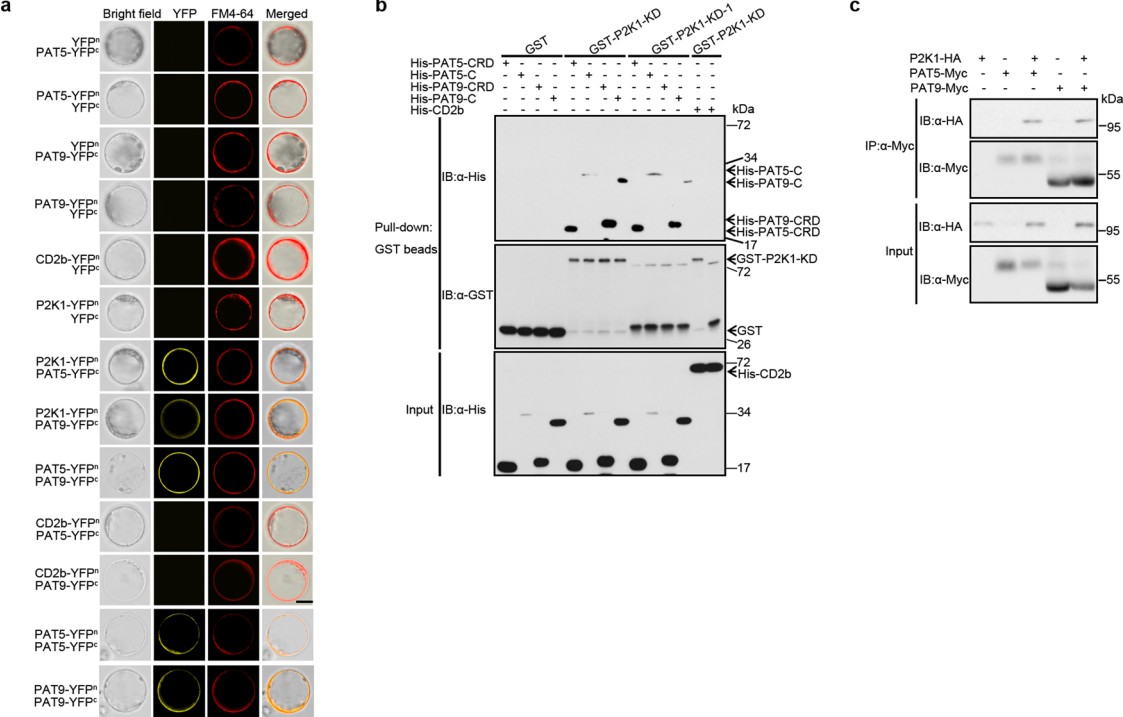

**Fig. 2 PAT5 and PAT9 directly interact with P2K1 in vitro and in vivo. a** Interactions of PATs and P2K1 receptor at *Arabidopsis* protoplast plasma membrane. FM4-64 was used to stain the plasma membrane. Bar = 20 μm. **b** Directly interaction of PATs and P2K1 in vitro. Purified, recombinant proteins were incubated with Glutathione Sepharose 4B beads followed by GST-His pull-down assay. His-CD2b (At5g09390, CD2-binding protein-related, a substrate of P2K1 from KiC assay data) was used as a negative control. **c** PAT5 and PAT9 interact with P2K1 in vivo. The indicated constructs were transiently expressed in *Arabidopsis* wild-type protoplasts followed by Co-IP assays. All experiments were performed and analyzed three times with similar results.

of assays to look for protein-protein interaction. For example, split-luciferase complementation imaging (LCI) assays in *Nicotiana benthaminana* showed that both PAT5 and PAT9 can interact in vivo with unstimulated P2K1, but not RBOHC (Supplementary Fig. 2c). Similarly, bimolecular fluorescence complementation (BiFC) assays in *Arabidopsis* protoplasts were used to specifically test for interactions at the PM. The yellow fluorescence signals were produced in the combination of PAT5/9-YFP$^C$ with P2K1-YFP$^N$ and merged well with the PM marker FM4-64 signal (Fig. 2a). Interestingly, PAT5 and PAT9 appeared to form heterodimers and homodimers in both assays (Fig. 2a and Supplementary Fig. 2c). These results suggest that DAMP treatment may enhance the association between PAT5/PAT9 and P2K1 on the PM.

The LCI and BiFC results show that PAT5/PAT9 and receptors can form a complex, but do not indicate which domains are responsible for their interaction. Therefore, we performed in vitro pull-down assays using GST- or His-tagged recombinant proteins purified from *E. coli*. The PAT5 and PAT9 proteins are composed of four transmembrane domains (TMDs), the catalytic DHHC-Cysteine rich domain (DHHC-CRD) which is present in the cytosol and necessary for auto-acylation and for the modification of target proteins, and the N- and C-terminal cytoplasmic domains (Supplementary Fig. 2d). Remarkably, PAT5-CRD, PAT5-C, PAT9-CRD, and PAT9-C were able to directly bind to C-terminal cytoplasmic domain P2K1-KD (Fig. 2b). Meanwhile, an interaction with the kinase inactive form of the P2K1 kinase domain (P2K1-KD-1) was also observed in this pull-down assay (Fig. 2b), suggesting that PAT5/PAT9-P2K1 interactions are independent of receptor autophosphorylation. Moreover, the PAT5-P2K1 and PAT9-P2K1 interactions were also detected in co-immunoprecipitation (Co-IP) assays when PAT5-Myc or

PAT9-Myc were co-expressed with P2K1-HA in wild-type protoplasts (Fig. 2c). Taken together, these various assays indicate that PAT5 and PAT9 directly interact with P2K1 *in planta*.

**P2K1 directly phosphorylates and stimulates PAT5 and PAT9.** We next sought to determine whether PAT5 and PAT9 are the direct phosphorylation substrates of P2K1 using kinase assays. Consistent with the KiC assay results, P2K1-KD strongly trans-phosphorylated PAT5-CRD, PAT5-C, PAT9-CRD, and PAT9-C, whereas the kinase dead P2K1-KD-1 failed to phosphorylate them (Fig. 3a), indicating that P2K1 can directly phosphorylate PAT5 and PAT9.

To further explore the role of P2K1 in PAT9 phosphorylation during PTI in planta, we investigated the key phosphosites found by the KiC assay. We found that the phosphosites of PAT9 in the KiC assay were conserved in PAT9 orthologs or paralogs in different species, such as soybean, rice, and maize (Supplementary Fig. 2e). We amplified the full-length genomic DNA of PAT9 with the native promoter, and substituted Thr107 and Ser109 with Ala (T107A/S109A), which blocks phosphorylation, or Asp (T107D/S109D), which mimics phosphorylation, and then transformed these constructs into *pat9* mutant plants (Supplementary Fig. 2d, f). The ATP-elicited cytoplasmic $Ca^{2+}$ influx and ROS production in the T107A/S109A PAT9 variant plants were similar to the *pat9* mutant plants and significantly higher compared to plants transformed with the WT PAT9 (Fig. 3b, c), demonstrating that Thr107 and Ser109 are required for the activation of PAT9. Conversely, phospho-mimic mutations of T107 and S109 (T107D/S109D) were remarkably reduced in their ability to stimulate the ATP-elicited $Ca^{2+}$ influx and ROS production (Fig. 3b, c). To confirm the important role of these two phosphosites in mediating ATP-induced bacterial defense, we

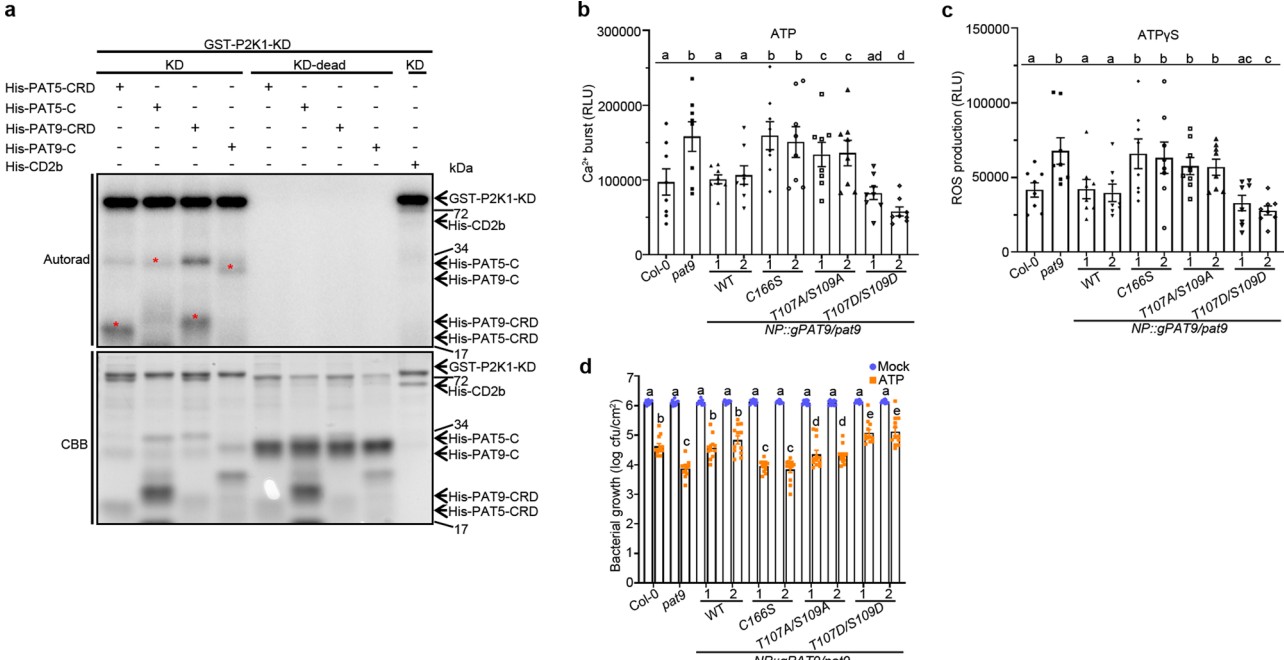

**Fig. 3 P2K1 phosphorylates PAT5 and PAT9 to regulate immune responses. a** P2K1 directly phosphorylates PAT5 and PAT9. Purified P2K1 kinase domain recombinant proteins were incubated with PAT5/PAT9-CRD and -C domains in an in vitro kinase assay. Autophosphorylation and transphosphorylation were measured by incorporation of γ-[$^{32}$P]-ATP. MBP and GST-CD2b were used as positive and negative controls, respectively. The protein loading was measured by CBB staining. Red stars represent the trans-phosphorylated proteins. Experiments were repeated two times with similar results. **b, c** Phosphorylation of PAT9 regulates ATP-induced calcium influx and ROS production. The indicated genomic *PAT9* variants were transformed into the *pat9* mutant, and then examined for ATP-induced calcium influx and ROS production for 30 min. RLU relative luminescence units. Error bars indicate ±SEM; $n =$ 8 (biological replicates); means with different letters are significantly different ($p < 0.05$); *P*-values indicate significance relative to Col-0 and were determined by one-sided ANOVA with multiple comparisons and adjusted using Benjamini–Hochberg post-test. **d** PAT9 Phosphorylation mediates ATP-induced restriction of bacterial growth. The indicated plant leaves were syringe-infiltrated with $10^6$ cfu/mL$^{-1}$ of *Pst*. DC3000 after 24 h water (mock) or 400 μM ATP treatment, which is different from surface inoculation (Fig. 1e). Bacterial numbers were determined 3 days post inoculation. Error bars indicate ±SEM; $n = 12$ (biological replicates); means with different letters are significantly different ($p < 0.05$); *P*-values indicate significance relative to Col-0 with mock treatment and were determined by one-sided ANOVA with multiple comparisons and adjusted using Benjamini–Hochberg post-test.

pretreated mutant leaves with water (mock) or 400 μM ATP for 24 h, and then syringe-infiltrated with $10^6$ cfu/mL$^{-1}$ of *Pst*. DC3000, which is different from the surface inoculation (Fig. 1e), and then quantified bacterial numbers 3 days post inoculation. Relative to wild-type plants, the T107A/S109A variants showed greater resistance to bacterial growth, while the T107D/S109D phospho-mimic plants were more susceptible (Fig. 3d). Based on these data, we proposed that T107 and S109 phosphosite variants appear to be partially functional. Meanwhile, to verify whether the key residue DHHC catalytic domain is responsible for PAT9 function, we mutated this sequence to DHHS (C166S) and expressed this variant in the *pat9* mutant plants. The C166S PAT9 variant plants showed the same Ca$^{2+}$ influx, ROS production and bacterial defense as the *pat9* mutant (Fig. 3b, d), demonstrating that the DHHC domain is required for the full function of the PAT9 protein.

The above data, together with PTI-induced phosphorylation of the PAT5 and PAT9 proteins, indicate that P2K1 can directly phosphorylate and activate PAT5 and PAT9 in an ATP-induced manner, resulting in dampening of the ATP-induced PTI response including the rapid elevation of cytoplasmic Ca$^{2+}$ and ROS production, and restriction of bacterial invasion.

**P2K1 is *S*-acylated by PAT5 and PAT9.** To investigate whether P2K1 can be *S*-acylated and which residues are targeted, we first predicted the potential *S*-acylation residues using GPS-lipid 1.0 software (online http://lipid.biocuckoo.org/webserver.php;

Supplementary Fig. 3). Based on these predictions, we generated a series of mutant forms of P2K1 in which cysteine (C) sites were individually mutated to serine (S). These mutant proteins were then transiently expressed in *Arabidopsis* protoplasts and their *S*-acylation status determined through a parallel *S*-acylation assay with or without the hydroxylamine thioester-cleavage step. Compared with the wild-type P2K1 form, the levels of *S*-acylation of single mutants such as P2K1$^{C394S}$ and P2K1$^{C407S}$, but not P2K1$^{C538S}$ and P2K1$^{C559S}$, were significantly reduced, while *S*-acylation of the double mutant P2K1$^{C394/407S}$ was below the level of detection (Supplementary Fig. 4a). Remarkably, P2K1$^{C394/407S}$ protein migrated faster than the wild-type P2K1 when separated by SDS-PAGE (Supplementary Fig. 4a). The substitution of C residue with a S residue could provide a potential phosphorylation site, and it is difficult to distinguish from phosphorylated P2K1. Therefore, we also replaced each C residue with alanine and transformed into *p2k1-3* mutant plants (Supplementary Fig. 4e). Consistent with C to S mutants, the levels of *S*-acylation of P2K1$^{C394A}$, P2K1$^{C407A}$ and P2K1$^{C394/407A}$ were remarkably reduced (Fig. 4a). Furthermore, loss of PAT5 and PAT9 function reduced P2K1 *S*-acylation levels, while complementation by expression of the wild-type PAT5 or PAT9 proteins restored these deficiencies (Fig. 4b). However, the PAT5$^{C188S}$ and PAT9$^{C166S}$ mutants, in which the DHHC catalytic domain was mutated to DHHS, lost auto-*S*-acylation activity and failed to rescue P2K1 *S*-acylation levels (Fig. 4b) when expressed in the *pat* mutant background. The above data indicate that PAT5 and PAT9 *S*-acylate P2K1 *in planta*.

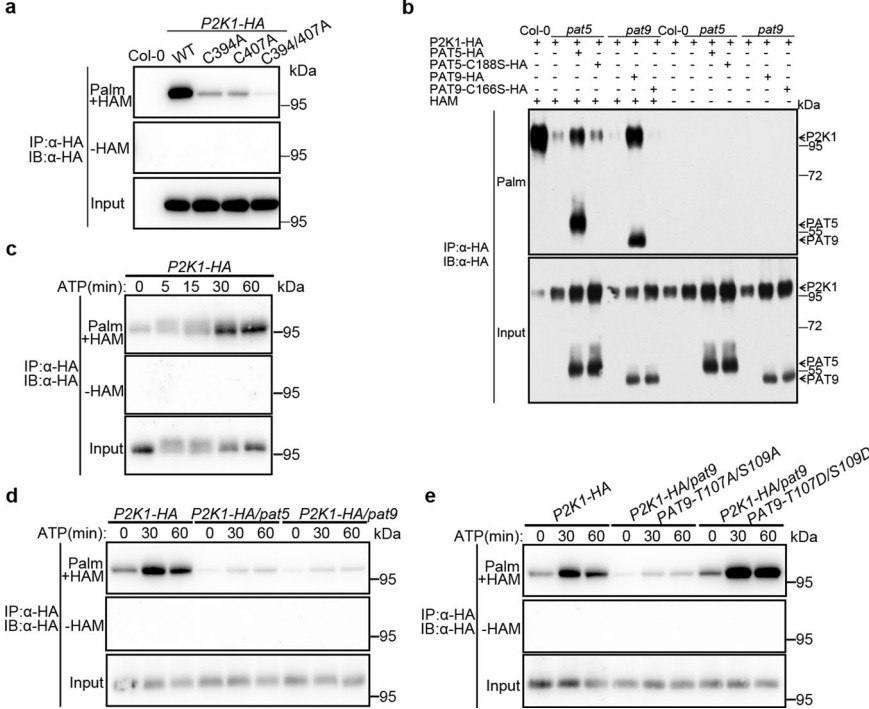

**Fig. 4 PAT5 and PAT9 S-acylate P2K1. a** Detection of P2K1 S-acylation and their associated residues in the stable *P2K1-HA* transgenic plants. The S-acylation levels were detected by an acyl-resin capture (acyl-RAC) assay. Palm- palmitoylated protein, Depalm- depalmitoylated protein. HAM, hydroxylamine, for cleavage of the Cys-palmitoyl thioester linkages. Experiments were repeated three times. **b** PAT5 and PAT9 S-acylate P2K1 *in planta*. Protoplasts of *Arabidopsis* wild-type Col-0, *Atpat5* and *Atpat9* mutants were transfected with plasmids as indicated, followed by acyl-RAC assays. Experiments were repeated three times. **c–e** Dynamic S-acylation levels of P2K1 in stable transgenic plants upon eATP addition. The *P2K1-HA* transgenic plant was crossed with *pat5*, *pat9*, *NP::PAT9-T107/AS109A/pat9*, and *NP::PAT9-T107D/S109D/pat9* plants and then infiltrated with 200 μM ATP. All experiments were repeated and analyzed three times with similar results.

In order to better examine the dynamic effects of S-acylation on the P2K1 receptor, we treated the stable transgenic *P2K1-HA* plant with eATP from 5 to 60 min. Under these conditions, S-acylation of P2K1-HA was remarkably increased 15 and 30 min after elicitation (Fig. 4c). To investigate the relationship between S-acylation and phosphorylation of the P2K1 receptor, we treated plants with MG132, an inhibitor of proteasome-mediated protein degradation. Interestingly, we found that the dynamics of P2K1 S-acylation showed an inverse relationship to P2K1 autophosphorylation (Supplementary Fig. 4b). Furthermore, the eATP-induced P2K1 S-acylation did not increase in the *pat5* or *pat9* mutants background (Fig. 4d), revealing that PAT5 and PAT9 are required for the eATP-elicited P2K1 S-acylation. Additionally, the phospho-dead PAT9 variant PAT9-T107AS109A plants failed to complement the eATP-induced P2K1 S-acylation, but phosphomimetic PAT9 variant PAT9T107DS109D plants showed stronger P2K1 S-acylation than wild type (Fig. 4e). These results, together with above data, show that PAT5 and PAT9 S-acylate P2K1 in an ATP-induced and time-dependent manner, and P2K1 S-acylation requires phosphorylation of PAT5/9.

Another important function of S-acylation is as a mediator of subcellular trafficking or membrane association, so we examined whether S-acylation regulates the localization of P2K1. First, we expressed GFP- and RFP-tagged fluorescent proteins in *Arabidopsis* protoplasts, the P2K1C394407S showed an identical PM localization as wild-type P2K1 (Supplementary Fig. 4c), indicating that P2K1 is correctly trafficked to the plasma membrane in the absence of S-acylation. We also crossed *35S:P2K1-GFP* plants with *pat5* and *pat9* mutant plants and found that the PM localization of P2K1 was not affected in both mutant backgrounds (Supplementary Fig. 4d), suggesting that S-acylation is not involved in the subcellular targeting of the P2K1 receptor.

**S-acylation mediates P2K1 phosphorylation and turnover in a ligand-dependent manner.** Given that PAT5 and PAT9 directly interact with, are phosphorylated by and S-acylate P2K1 in a DAMP-dependent manner, we sought to determine whether the activation of P2K1 is regulated by PAT5 or PAT9. Previous results showed that autophosphorylation of P2K1 is induced by ATP elicitation[19]. Therefore, we expressed P2K1-HA in the *Atpat9* and *Atpat5* mutant background and found that, compared with *P2K1-HA* expressed in wild-type plants, the *P2K1-HA/Atpat9* and *P2K1-HA/Atpat5* plants showed greater P2K1 phosphorylation at 5, 15, and 30 min after ATP treatment (Fig. 5a). The mobility of the phosphorylated protein was shifted by treatment with lambda protein phosphatase (lambda PP) to release any phosphate groups. The ubiquitin-mediated degradation of FLS2 is ligand, dose, and time dependent[41]. Similarly, the P2K1 protein was reduced to ~17% with 400 μM ATP treatment for 1 h, and this receptor turnover was also time dependent (Fig. 5a, b, d). Moreover, loss of PAT9 or PAT5 in *P2K1-HA/pat9* or *P2K1-HA/pat5* plants significantly inhibited this ATP-induced P2K1 turnover (Fig. 5a). On the other hand, the addition of 2-bromopalmitate (2-BP), a palmitate analog known to inhibit S-acyl transferase activity, increased P2K1 accumulation but not phosphorylation (Supplementary Fig. 5a). These data demonstrate that PAT5 and PAT9 modulate activation of P2K1 protein in a ligand-induced manner, mediated by effects on both autophosphorylation and subsequent protein turnover.

In order to better understand how S-acylation mediates receptor activity upon elicitor stimulation, we examined mutated P2K1 protein in *Arabidopsis* with the substitution of C residue with alanine, but not a S residue, which may provide a potential phosphorylation site (Supplementary Fig. 4a). Expression of the

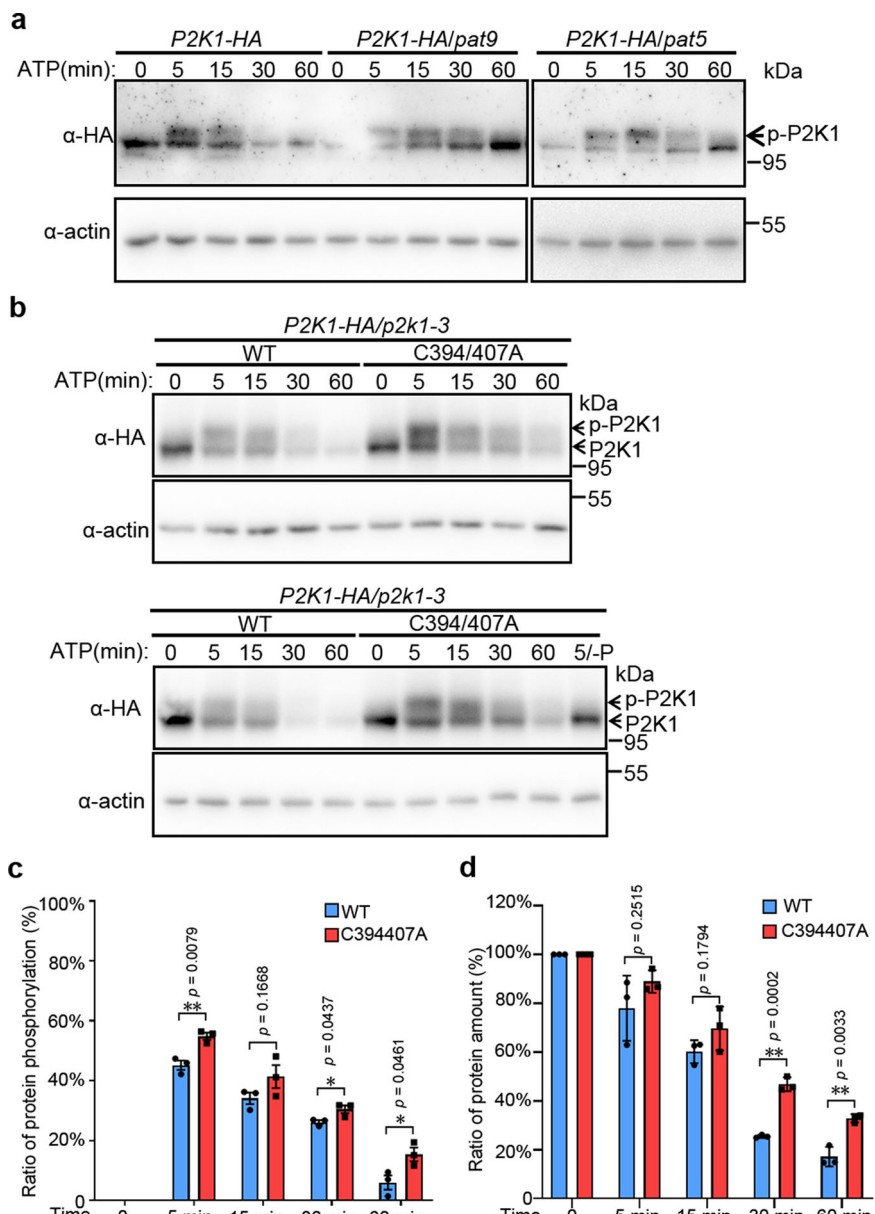

**Fig. 5 PAT9 regulates ATP-induced P2K1 phosphorylation and turnover. a** PAT5/9 controls P2K1 phosphorylation and turnover. P2K1-HA was analyzed by immunoblot in wild type and *Atpat5* and *Atpat9* mutants background upon addition of 400 μM ATP. p-P2K1 phosphorylation of P2K1. Actin protein was used as loading control. Experiments were repeated three times with similar results. **b–d** Analysis of turnover and phosphorylation of P2K1 proteins modified (C → A) at the site of *S*-acylation. The indicated constructs were expressed into *Arabidopsis p2k1-3* mutant plant and treated with 400 μM ATP. Lambda PP (-P) was added to release phosphate groups. In panel **c** and **d**, the ratio of P2K1 protein phosphorylation and total protein amount were measured from panel **b** and another replicate with Image J. Error bars indicate ±SE; $n = 3$ (biological replicates). *$p < 0.05$, **$p < 0.01$, *P*-values indicate significance relative to WT and were determined by one-sided ANOVA with unpaired, two-tailed Student's *t* test.

P2K1$^{C394A}$ and P2K1$^{C407A}$ protein showed more P2K1 protein phosphorylation but less protein turnover than the wild-type form upon ATP addition (Supplementary Fig. 5b). Meanwhile, the double substitute forms of P2K1$^{C394/407A}$ protein also showed a significant reduction in ligand-induced protein turnover from 47% of wild-type levels remaining (26% in WT form) at 30 min to 33% (17% in WT form) at 1 h (Fig. 5b, d and Supplementary Fig. 5c). Furthermore, ATP-induced P2K1 protein phosphorylation increased to 45% at 5 min and then decreased to 6% at 1 h (Fig. 5a–c). Meanwhile, the P2K1$^{C394/407A}$ protein phosphorylation was remarkably increased relative to the wild-type P2K1 protein from 55% (45% in WT) at 5 min to 15% (6% in WT) at 1 h (Fig. 5b, c and Supplementary Fig. 5c). In summary, these

results further demonstrate that *S*-acylation of key residues of the P2K1 protein plays an important role in mediating ligand-induced autophosphorylation and protein turnover.

**P2K1 *S*-acylation regulates PTI**. In order to demonstrate the biological relevance of P2K1 *S*-acylation mediated by PAT5 and PAT9, we generated stable transgenic plants expressing the mutated *S*-acylation site proteins in the *p2k1-3* mutant background using the full-length genomic sequences with a 1.5-kb native promoter (Supplementary Fig. 5d). Similar to the data derived using the *Atpat5* and *Atpat9* mutant plants, ligand-induced elevation of cytoplasmic calcium levels in plants

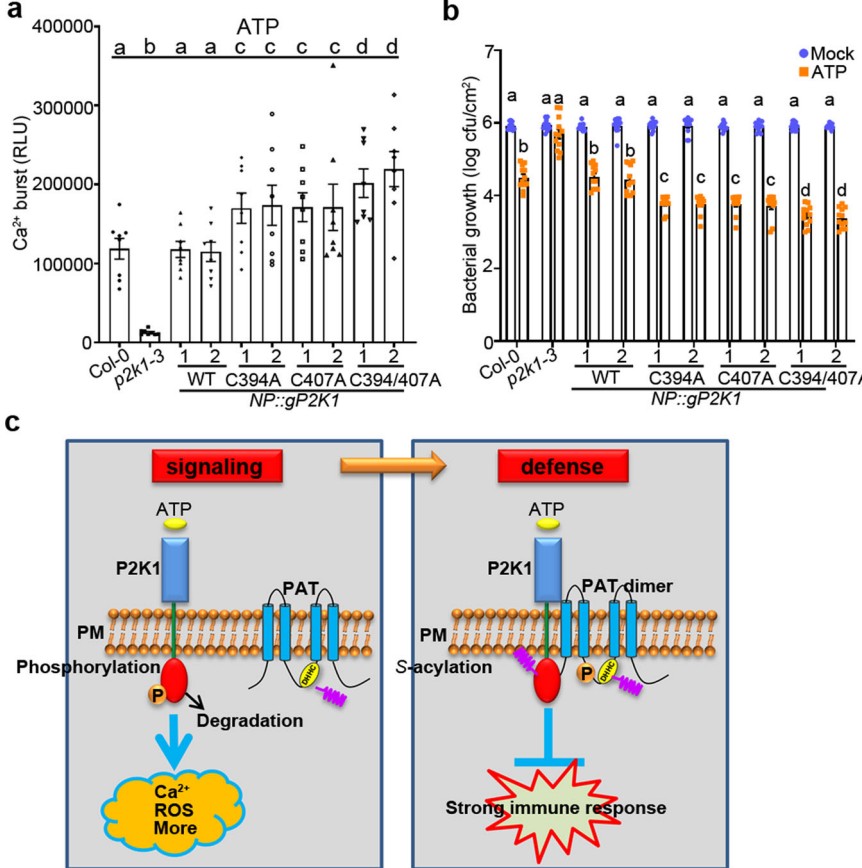

**Fig. 6 S-acylation of P2K1 negatively regulates PTI. a, b** S-acylation of P2K1 mediates ATP-induced calcium influx and bacterial defense. The indicated 5-day-old seedlings were treated with 100 μM ATP for calcium influx in 15 min. For bacterial growth, the indicated plant leaves were pretreated with 400 μM ATP for 24 h, and then inoculated with Pst. DC3000 and bacterial growth measured by plate counting after 3 days post inoculation. Values represent the mean ± SEM, $n = 8$ in **a**, $n = 12$ in **b** (biological replicates); Means with different letters are significantly different ($p < 0.05$); P-values indicate significance relative to Col-0 with mock treatment and were determined by one-sided ANOVA with multiple comparisons and adjusted using Benjamini–Hochberg post-test. **c** Model for the role of P2K1 receptor S-acylation in eATP signaling. Upon addition of the activating ligand eATP, the P2K1 receptor is rapidly autophosphorylation and phosphorylates downstream targets, leading to PTI immune response and P2K1 protein turnover. P2K1 directly interacts with and phosphorylates PATs to activate PATs S-acylation upon ATP treatment. Activation of PATs S-acylates P2K1 and then inactivates P2K1 phosphorylation and turnover, following dampens the immune response to protect plant growth.

expressing P2K1$^{C394A}$, P2K1$^{C407A}$ or P2K1$^{C394/407A}$ variants were significantly higher than similar mutant plants complemented with the wild-type P2K1 (Fig. 6a). The ATP-induced resistance to the virulent bacterium Pst. DC3000 was also significantly enhanced in plants expressing P2K1$^{C394A}$, P2K1$^{C407A}$, or P2K1$^{C394/407A}$ compared to the corresponding wild-type controls (Fig. 6b). In summary, consistent with both the in vivo and in vitro biochemical measurements, S-acylation of P2K1 negatively regulates the elicitor mediated PTI responses.

## Discussion

Unlike animals, plants are unable to move in response to danger or stress nor do they possess humoral and cellular immunity. Instead, plants have evolved an extensive and complex system of receptors that recognize pathogens, pests, and cellular damage through a large and diverse family of PM-localized RLKs. Ligand activation of these receptors leads to autophosphorylation, transphosphorylation of downstream proteins, and other cellular events culminating ultimately into a robust innate immunity response[42]. Concomitant with these events, the receptor is rapidly turned over through endocytosis, presumably to dampen the immune response that, while protective, is also detrimental to

overall plant growth. Although each PRR recognizes a specific PAMP/DAMP signal, in general, the downstream signaling responses are highly similar regardless of which PRR is activated.

Dynamic S-acylation determines protein function by influencing their association with membranes, compartmentalization in membrane domains, trafficking, and stability[23]. Our current study demonstrates that plant PRR activation is mediated by specific PAT proteins on the PM through S-acylation. Ligand activation of the P2K1 receptor leads to rapid autophosphorylation, downstream signaling, receptor turnover and, within 30 min restoration of S-acylation. Importantly, ATP-induced P2K1 S-acylation shows a clear inverse relationship with P2K1 autophosphorylation. Our hypothesis is that, S-acylation acts as a negative regulator of innate immunity, insuring a steady-state inactive state for P2K1 that insures against spurious activation, which might be detrimental to growth. Consistent with this model (Fig. 6d), treatment with inhibitors of S-acylation or studies using mutants defective in PAT activity (pat5/pat9) or receptors mutated in their specific S-acylation sites all led to a marked elevation of the innate immunity response whether measured using cellular assays (e.g., cytoplasmic calcium levels) or pathogen sensitivity. A similar dynamic of receptor post-translational modification also occurs during activation of the

epidermal growth factor receptor (EGFR) in human tumorigenesis and cancer progression[28]. Hence, at least for these specific examples, the 'yin-yang' between *S*-acylation and phosphorylation/turnover is common across plants and animals.

Previous phylogenetic analysis revealed that the *Arabidopsis* PAT proteins were mainly clustered in three clades[29]. PAT5 and PAT9 belong to the largest and most homogenous group, and show sequence conservation with PAT6, PAT7, and PAT8. Our study reveals that PAT5 and PAT9 negatively control plant innate immunity mediated by PAMP/DAMP, and are, at least partially, functionally redundant. Meanwhile, we find that the claimed phosphosites of PAT9 are highly conserved in PAT6 and PAT8 in *Arabidopsis*, or orthologous genes in rice, soybean, and maize (Supplementary Fig. 2e). Based on these data, we propose that PAT5/6/7/8/9 clade may be functionally redundant and similar mechanisms could apply to species other than *Arabidopsis*.

In summary (Fig. 6d), under steady-state conditions, the plant PRR P2K1 is *S*-acylated and distributed into the PM in an inactive state. In response to extracellular ATP binding, the P2K1 receptor was deacylated, which occurs concomitantly with P2K1 autophosphorylation. This activation ultimately leads to receptor endocytosis and ubiquitin-mediated protein turnover. PRR activation triggers downstream innate immune signaling, including rapid elevation of calcium influx, ROS, MAPKs activation, and other events leading to a robust innate immunity response. Meanwhile, activated P2K1 directly phosphorylates and stimulates PAT5 and PAT9, which will then *S*-acylate P2K1 receptor through the DHHC catalytic domain. In the case where the P2K1 receptor is not degraded, *S*-acylation is restored as an alternative means to dampen the immune response. These results are important since they clearly demonstrate how *S*-acylation mediates plasma membrane machinery and signaling during plant innate immunity, drawing clear parallels with well-studied signaling systems in animals. Missing steps include identification of the specific thioesterases that mediate the rapid deacylation and a clearer understanding of the full dynamics of the complex interactions within the receptor complex at the PM. Hence, much remains to be discovered about how plants, as well as other organisms, are able to survive in the face of the various pathogens or increasing severity of environmental change through the PM receptor complex.

## Methods

**Plant materials and growth conditions**. All *Arabidopsis thaliana* plants used in this study are derived from the Columbia (Col-0) ecotype and express an T-DNA carrying aequorin[43], including *p2k1-3* (Salk-042209), *pat5* (GABI-322D08), *pat9* (SALK-003020C), and *pat9-2* (SALK_206051C). Additional transgenic lines in this study are described below. Plants were grown in soil or 1/2 MS medium containing 1% sucrose at 21–23 °C, 60–70% humidity in a growth chamber under long day (16 h light/8 h dark) conditions.

**Constructs and transgenic plants**. Full-length CDS or genomic DNA of P2K1 (At5g60300), PAT5 (At3g48760), PAT9 (At5g50020), RBOHC (AT5G51060), and CD2b (At5g09390), as well as their kinase domain or C-terminal domain, were amplified from wild-type plants using gene-specific primers (Supplementary Table 1). The PCR products were cloned into pDONR-Zeo or pGEM-T Easy vectors. The different mutant forms were generated by PCR-mediated site directed mutagenesis.

In order to generate constructs for the LCI assays in tobacco leaves, the full-length DNA from the pDONR-Zeo vector were cloned into pCAMBIA1300-Nluc and pCAMBIA1300-Cluc using LR cloning. For BiFC assays, we used pAM-PAT-35SS::YFP:GW, pAM-PAT-35SS::YFPc:GW, and pAM-PAT-35SS::YFPn:GW[44] as the destination vectors to form fusions with split YFP at the C-termini of proteins in *Arabidopsis* protoplasts using LR cloning.

To express specific proteins in *Arabidopsis* protoplasts, different mutated forms of P2K1, PAT5, and PAT9 from the source pDONR-Zeo source vectors were cloned into pUC-GW14 and pUC-GW17[19] vectors using LR cloning.

In order to generate *NP::ATPAT5/pat5* and *NP::ATPAT9/pat9* stable complemented transgenic plants or different mutated variants, the coding sequences driven by their native promoters were PCR-amplified from wild-type

genomic DNA or CDS from cDNA, and then cloned into pGWB1, pGWB5 and pGWB13 using LR cloning.

In order to generate constructs used for expressing different mutant forms of P2K1, the full-length P2K1 genomic sequence including ~1.5 kb native promoter was amplified from genomic DNA by PCR, and cloned into pGWB1, pGWB13, pGWB16[45] using LR cloning, respectively.

For constructs expressing recombinant proteins in *E. coli*, the DNA fragments of P2K1-KD cut with EcoRI and XhoI was inserted into pGEX-5X-1. To gain His-tagged constructs, DNA fragments of PAT5-CRD, PAT5-C, PAT9-CRD, and PAT9-C cut with BamHI and XhoI were cloned into pET28a, while CD2b using SacI and XhoI, respectively.

**Kinase client assay (KiC assay)**. The KiC assay, instrument and detailed search parameters were used as before[19,40]. More than 2100 peptides identified from phosphorylation sites taken from a number of studies were individually synthesized and then incubated with the purified, recombinant GST-DORN1-KD kinase domain followed by ATP addition. The peptide mixture was then analyzed using a Finnigan Surveyor liquid chromatography (LC) system attached to a LTQ Orbitrap XL ETD mass spectrometer. Two sets of empty vectors (GST and MBP) and two kinase-dead proteins, GST-P2K1-1 (D572N) and GST-P2K1-2 (D525N), were used as negative controls.

**Calcium influx assay**. Briefly, 5-day-old seedlings were individually incubated with 50 μl of reconstitution buffer containing 10 μM coelenterazine (Nanolight Technology, Pinetop, AZ), 2 mM MES buffer (pH 5.7), and 10 mM CaCl$_2$ in the wells of a 96-well plate in the dark at room temperature overnight. The next morning, 50 μl of treatment solution (concentration was double strength to give a set final concentration of 25 mM MES and 100 μM ATP (Sigma, A2383), 1 μM flg22 (GenScript), or 50 μg/ml chitin (Sigma, C9752)) was added to each well, and the luminescence was monitored using a CCD camera (Photek 216; Photek, Ltd.). The luminescence was monitored about 15 min for ATP treatment, 30 min for flg22 and chitin treatments.

**Oxidative (ROS) burst assay**. Leaf disks were taken from 4- or 5-week-old plants and incubated with 50 μl ddH$_2$O in the wells of a 96-well plate in the dark overnight. The next day, 50 μl 2x chemiluminescent luminol buffer was added to each well to be a final concentration of 25 μM luminol, 20 μg/ml horseradish peroxidase and 200 μM ATPγS (Sigma, A1388), 1 μM flg22, or 50 μg/ml chitin. Luminescence was immediately monitored using a CCD camera (Photek 216; Photek, Ltd.) for 30 min.

**MAPK phosphorylation assay**. Leaf disks from 4- or 5-week-old plants were incubated in the ddH$_2$O overnight at 23 °C, and then treated with 200 μM ATPγS, 1 μM flg22, or 100 μg/ml chitin for 0, 15, 30, or 60 min. Total protein was extracted with extraction buffer containing 50 mM Tris (pH 7.5), 150 mM NaCl, 0.5% Triton-X 100, and 1x protease inhibitor for 30 min on ice. The extracted total proteins were separated by 10% (w/v) SDS-PAGE gel and detected by immunoblotting with anti-phospho-p44/p42 MAPK antibody (Cell Signaling, 9101, dilution, 1:1000).

**Bacterial growth assays**. Bacterial growth was performed using flood inoculation of seedlings[46]. Generally, 50 ml of *P. syringae pv. tomato* DC3000 ($5 \times 10^6$ colony-forming units (CFU) ml$^{-1}$) bacterial suspension containing 10 mM MES pH 5.7, 10 mM MgCl$_2$, 0.025% Silwet L-77 was dispensed into plates containing about 14-day-old seedlings for 2–3 min. For, ATP-induced bacteria invasion, 4- or 5-week-old plant leaves were pretreated with 400 μM ATP, 10 mM MES pH 5.7, for 24 h, and then syringe-infiltrated with $10^6$ cfu/mL of *Pst*. DC3000. After 3 days post inoculation[47], either the leaves or the whole seedling without the root were ground in 10 mM MgCl$_2$, diluted serially, and plated on LB agar with 25 mM rifampicin. Colonies (CFU) were counted after incubation at 28 °C for 2–3 days.

**Gene expression (qRT-PCR)**. RNA was extracted from seedlings or leaves using TRIzol Reagent (Invitroge) and the first-strand cDNA synthesis using M-MLV RT (Promega) according to the manufacturer's instructions. qRT-PCR was carried on an Applied Biosystems QuantStudio 6 Flex (ABI). The relative gene expression was quantified using ΔΔCt method and normalized to UBQ expression. The primers used in the study are in Supplementary Table 1.

**Split-luciferase complementation imaging assay**. The *Agrobacterium tumefaciens* (GV3101) cells containing the indicated constructs were infiltrated into 4-week-old leaves of *N. benthamiana* and then incubated at room temperature for 48 h before LUC activity measurement. In all, 1 mM D-luciferin was sprayed onto the leaves and then kept in the dark for 5–10 min to allow the chlorophyll luminescence to decay, the luminescence was monitored using a CCD camera (Photek 216; Photek, Ltd.).

**Arabidopsis protoplast isolation and transformation**. The isolation and transfection of Arabidopsis protoplasts were performed as previously described[19] and used for the various assays as indicated. Briefly, 2 g of 14 days old seedlings were sliced and mixed with 15 ml TVL Solution (0.3 M sorbitol and 50 mM CaCl$_2$). In all, 20 ml of Enzyme Solution was then added containing 0.5 M sucrose, 10 mM MES-KOH pH 5.7, 20 mM CaCl$_2$, 40 mM KCl, 1% Cellulase (Onozuka R-10), and 1% Macerozyme (R10). This enzyme solution containing protoplasts were filtered through a 75-mm nylon mesh after a gentle swirling motion at room temperature for 15–18 h. Next, the protoplasts were gently covered with 10 ml W5 Solution (2 mM MES pH 5.7, 154 mM NaCl, 125 mM CaCl$_2$, and 5 mM KCl) without disturbing the sugar content gradient, following centrifugation at 100×g for 7 min. About 10 ml of protoplasts were collected at the interface of Enzyme Solution and W5 Solution were transferred to a new tube. The protoplasts were then washed twice with 15 ml of W5 Solution and centrifuged for 5 min at 60 g. The pelleted protoplasts were resuspended in 1–3 ml MMG Solution containing 4 mM MES pH 5.7, 0.4 M mannitol and 15 mM MgCl$_2$. For DNA-PEG–calcium transfection, 10–20 μg (about 10 μl) of plasmid was added to 100 μl protoplasts and mixed gently. An aliquot of 110 μl of PEG solution was mixed with this DNA-protoplasts by gently tapping the tube, and then incubated at room temperature for 15 min. The transfection mixture was mixed with 450 μl W5 solution to stop the transfection process, and centrifugated at 100×g for 1–2 min. The pelleted protoplasts were resuspended with 1 ml W5 solution. After incubation overnight in the dark at 23 °C, the protoplast solution was used for the various assays.

**Bimolecular fluorescence complementation assay**. N- and C-terminal YFP protein fusions plasmids were co-transformed into Arabidopsis protoplasts as described above and then incubated at 23 °C in a growth chamber overnight at dark. The YFP fluorescence was observed using a Leica DM 5500B Compound Microscope with Leica DFC290 Color Digital Camera. For FM4-64 staining, 2 μM FM4-64 was used and incubated for 5 min at room temperature (the plasma membrane stain).

**In vitro pull-down**. Recombinant proteins GST-P2K1-KD, GST-P2K1-KD-1, His-PAT5-CRD, His-PAT5-C, His-PAT9-CRD, His-PAT9-C, or His-CD2b were expressed in E. coli and affinity purified using Glutathione Resin (GenScript) and TALON® Metal Affinity Resin (Clontech), respectively. For pull-down, 5 μg GST and His recombinant proteins were incubated with 25 μl Glutathione Resin beads in the pull-down buffer containing 25 mM Tris-HCl pH 7.5, 100 mM NaCl, and 1 mM DTT for 2 h at 4 °C. The beads were washed more than seven times with the washing buffer 25 mM Tris-HCl pH 7.5, 100 mM NaCl, and 0.1% Triton-X 100. The bund proteins were eluted with 25 μl elution buffer containing 50 mM Tris-HCl pH 7.5–8, 15–20 mM GSH for ~15–30 min. The proteins were separated using SDS-PAGE gels and detected by immunoblotting using anti-His (Sigma, H1029, dilution, 1:1000) and anti-GST-Hrp (GenScript, A00130, dilution, 1:1000).

**In vitro phosphorylation assays**. For the in vitro kinase assay, 2 μg of purified GST-P2K1-KD or GST-P2K1-KD-1 kinases were incubated with 1 μg His-PAT5-CRD, His-PAT5-C, His-PAT9-CRD, His-PAT9-C, or His-CD2b as substrate in a 20-μl reaction buffer containing 50 mM Tris-HCl pH 7.5, 50 mM KCl, 10 mM MgCl$_2$, 10 mM ATP, and 0.25 μl radioactive [γ-$^{32}$P] ATP for 30 min at 30 °C. The reaction was stopped by 5 μl of 5x SDS loading buffer. The proteins were separated by SDS-PAGE (10%), followed by autoradiography for 3 h. The proteins within the gel were visualized by staining with Coomassie blue. Myelin basic protein (MBP) was used as a positive control.

**Co-immunoprecipitation assay**. Total proteins were extracted from protoplasts or plant tissues with an extraction buffer containing 50 mM Tris (pH 7.5), 150 mM NaCl, 0.5% Triton-X 100, and 1 × protease inhibitor for 1 h on ice. The samples were centrifuged at 20,000×g for 15 min at 4 °C, the supernatant was decanted and 1 μg anti-Myc or 30 μl anti-HA agarose was added to the supernatant and incubated for 4 h or overnight with end-over-end shaking at 4 °C. In all, 25 μl protein A resin was added for 2 h, spun down and washed seven times with extraction buffer. After washing, 25 μl 1x SDS-PAGE loading buffer was added and heated at 100 °C for 10 min. The proteins were separated by SDS-PAGE and detected by immunoblotting with anti-HA-HRP (Roche, 12013819001, dilution, 1:3000), anti-Myc-HRP (Sigma, SAB4700447, dilution, 1:3000).

**S-acylation assay**. The S-acylation assays were performed as previously described[48]. Generally, Arabidopsis protoplasts transfected with HA-tagged proteins were homogenized in lysis buffer containing 50 mM Tris (PH 7.5), 150 mM NaCl, 0.5% Triton-X 100, and 1x protease inhibitor for 1 h on ice. After centrifuged at 20,000×g for 15 min at 4 °C, 50 mM N-ethylmaleimide was added to the supernatant for blocking free sulfhydryl groups, and proteins were then immunoprecipitated using Anti-HA-Agarose beads with end-over-end shaking at 4 °C overnight. The next day, the beads were washed three times with lysis buffer and eluted in 100 μl 0.1 μg/μl HA peptide for 15 min. The eluted proteins were divided into two equal portions: one treated with 1 M hydroxylamine and the other with 1 M Tris-HCl (pH 7.4; as a control) in the presence of activated thiol-Sepharose 4B. Two hours later, the sepharose beads were washed three times with lysis buffer

without protease inhibitor at room temperature, and then resuspended in 1x protein loading buffer and heated at 100 °C for 10 min. Western blots were performed using anti-HA-HRP (dilution, 1:3000).

**Quantification and statistical analysis**. Statistical analysis was performed in GraphPad Prism 8. Error bars in the figures are standard deviation (SD) or the standard error of the mean (SEM = SD/(square root of sample size)), and number of replicates is reported in the figure legends. Statistical comparison among different samples was carried out by one-way ANOVA. Multiple comparison tests were corrected by controlling the false discovery rate (FDR) using Benjamini and Hochberg's method. Samples with statistically significant differences (*$p < 0.05$ or **$p < 0.01$ as indicated in the figure legends) were marked with different letters (a, b, c etc.).

**Reporting summary**. Further information on research design is available in the Nature Research Reporting Summary linked to this article.

## Data availability

All data supporting the findings of this study are included in this manuscript or further materials can be obtained from the corresponding author upon request. Source data are provided with this paper.

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

## Acknowledgements

Research reported in this publication was supported by the National Institute of General Medical Sciences of the National Institutes of Health (grant no. R01GM121445 to G.S.). The content is solely the responsibility of the authors and does not necessarily represent the official views of the National Institutes of Health. This work was also supported by the Next-Generation BioGreen 21 Program Systems and Synthetic Agrobiotech Center, Rural Development Administration, Republic of Korea (grant no. PJ01325403 to G.S.). Additional funding was provided within the framework of the 3rd call of the ERA-NET for Coordinating Action in Plant Sciences through NSF grant 1826803 (to G.S.). This work also supported by Chinese Universities Scientific Fund of China Agricultural University (No. 2020RC011 to D.C.), and the National Natural Science Foundation of China (32070285).

## Author contributions

D.C., F. H., and H.M. designed and performed the experiments and wrote the manuscript. N.A. and J.J.T. performed and supervised the kinase client screen for kinase targets. G.S. and D.C. supervised the study and edited the manuscript. All authors discussed the results and commented on the manuscript.

## Competing interests

The authors declare no competing interests.
