## [Peer Review File · Nature Communications]

REVIEWER COMMENTS

Reviewer #1 (Remarks to the Author):

S-Acylation of P2K1 mediates extracellular ATP-induced immune signaling in Arabidopsis

Dongqin Chen^{1,2}, Nagib Ahsan^{3,4}, Jay J. Thelen³ and Gary Stacey^{2,3,5*}

Reviewer comments

This article studied the role of S-acylation in the regulation of phosphorylation and degradation of the P2K1 receptor in response of elicitor stimulation in Arabidopsis. The experiments were well designed and carried out. The data were well presented with proper statistic analysis. However, I have the following comments:

1. Interaction between PAT5/9 and P2K1. The authors showed clearly that interactions between PAT5/9-CRD, PAT5/9-C with P2K1 were independent to the autophosphorylation status of P2K1 (Fig 2). Then P2K1-KD strongly trans-phosphorylate PAT5/9-CRD, PAT5/9-C, but P2K1-KD1 did not. However, the two phosphosites of PAT9 are at S106 and S107, outside of the CRD and C-ter regions of PAT9. If the interactions between PAT5/9 and P2K1 are S-acylation and phosphorylation dependent can the authors explain how?

2. P2K1 S-acylation. S-acylation of P2K1 by PAT5/9 and the cys sites of P2K1 for S-acylation were confirmed by the Acyl-RAC assay using transfected Arabidopsis protoplasts (Fig 4). Can the authors indicate if the PM localization of P2K1 CS variants were also changed? In addition, because stable transgenic lines for PAT5/9:pat5/9, PAT5/9CS:pat5/9 and P2K1:p2k1 and P2K1CS:p2k1 were generated by the authors, I would recommend further confirmation for P2K1 S-acylation and localization using these stable transgenic lines

3. Second T-DNA line for PAT5 and PAT9 should also be used

4. Line 200, 'directly phosphorylates and stimulates PAT5 and PAT9'.

You have only included the data for PAT9. Please also include data from PAT5

5. Localization of PAT5 and PAT9 have been published by Batisctic, 2016, therefore, it is not necessary to include your data. However, it would be good to include your localization data from your stable transgenic lines as this has not been done before.

Minor comments:

Various spelling mistakes, misuse of scientific terms, bad construction of a few sentences, missing references and sources of reagents were found throughout the m/s. The authors need to carefully edit the manuscript before re-submitting.

1. Methods. Please refer to relevant references for individual methods followed, such as the acyl-RAC, pAM-PAT-35SS:YFP:GW, pGWB13 etc etc. Sources of flg22, chitin, sources of various antibodies, etc

2. Line 180: please indicate conc and stained time as FM4-64 is NOT a PM specific dye and it only stains PM within the first few minutes before internalized

Line 76: FLS2, use full term

Line 80: 'lysin', change to lysine

Line 81: 'CERK1', full term please

Line 82: 'The first plant, due to cell/tissue damage', badly constructed sentence, please change

Line 120: 'Of the 24 encoded AtPATs, only sevenplay roles in....', please add 'early seedling growth and establishment (AtPAT15)'

Line 123: '....specific substrates for these AtPATs have not been identified ...'. This statement is not correct because CBL2,3 were identified as AtPAT10 substrates, for example (Zhou et al 2013).

Line 467&478:(PH 7.5)...., change to (pH 7.5)

Line 480: at 20,000 xg...., change to 20,000 xg (g should be italic)

Line 494: .. (*p<0.05 or **p<0.01....., p should be italic

Many more.....

Reviewer #2 (Remarks to the Author):

The manuscript titled "S-Acylation of P2K1 mediates extracellular ATP-induced immune signaling in Arabidopsis" by Chen et al. is a very interesting study on the role of PAT5 and PAT9 (protein S-acyl transferases) in the regulation of immune responses mediated by the eATP receptor P2K1 (and possibly other PRRs as well). The authors found that the receptor P2K1 is able to phosphorylate PAT5 and PAT9 and this phosphorylation is rapidly lost upon ligand detection by P2K1. PAT5 and PAT9 are shown to be able to s-acylate P2K1. This P2K1 acylation mediated by PAT5 and PAT9 is required for the dephosphorylation of P2K1 and its return to its inactive steady state. This study shows a novel understanding about the regulation of PRR mediated plant immune responses by protein S-acyl transferases. The manuscript is well written, and experiments have been logically designed to answer key questions. There are some revisions to be made to the manuscript that will help to convey its message clearly.

Major points:

1. The authors should include a discussion of the interesting redundancy of the PAT5/9 clade. A simple phylogenetic tree should be included. The weak phenotypes of the pat5/9 single mutants may be explained by the redundancy of the family. Also, heterodimerization is observed in the single mutant, providing an explanation that even single mutants would exhibit observable phenotypes. Of course, to confirm these, the authors need to create the pat5/6/7, pat8/9, and pat5/6/7/8/9 combinatorial mutants. The prediction would be that the 5/6/7 and 8/9 would show the same phenotype as 5/6/7/8/9 to support the heterodimerization model. As making these combinatorial mutants takes a long time, I would agree it is out of the scope of the current study.

2. In addition to the phylogeny, the authors should examine the existing microarray or RNA-seq datasets to examine the expression of the PAT5/9 clade. Are all 5 genes induced upon PTI/pathogen induction? Such analysis may help corroborate the whole story. The authors should also add the PAT dimerization in the model figure.

3. Figure 1

Panel d- include the double mutant pat5 pat9 in the assay

Panel f- make it clear that the 15-minute sample alone was treated with Lambda PP to show that the higher band seen is the phosphorylated form of the protein. The CBB control bands are not the same in the samples representing different times after ATP treatment. The experiment should be repeated such that the loading is consistent in the lanes.

Panel g- in addition to the mock treated sample without Lambda PP treatment there must also be a control with the addition of the phosphatase

Legend: the PAT9 complementation line has been incorrectly denoted as NP::ATPAT5-HA/Atpat5

4. Supplementary Figure 2A

Panel A

In lines 175-176 of the manuscript it is stated that "Interestingly, all these interactions were reduced 15 min after elicitation and then recovered 1 hour after elicitation (Supplementary Fig. 2a)."

In order to support this statement the 0 minute or prior to elicitation images of the *Nicotiana benthamiana* leaves need to be shown. Also, the figure should be organized better to make it easier to follow- the leaves showing the PAT5 and PAT9 interaction with P2K1 at different time points after elicitation (0, 15, 60 min post elicitation) could be placed together and the leaves showing other data such as PAT5 and PAT9's interaction with RBOHD and self and cross dimerization of the two PATs could be separate.

Lines 180-181: Interestingly, PAT5 and PAT9 appeared to form heterodimers and homodimers in both assays (Fig. 2a and Supplementary Fig. 2a)- There is no experiment shown in Figure 2A (BiFC) to show the formation of homodimers.

5. Figure 3

Panel A- the Coomassie stained blot is not very clean and there are multiple contaminating bands of sizes similar to those of the proteins under study which makes discerning the right bands

difficult. If possible, an anti-His blot can be performed which would give a much cleaner image of the input bands. Also, in the autoradiograph, in lanes 1 and 3 there is a band of size similar to that of His-PAT5-C. How do you know the band labelled as phosphorylated His-PAT5-C in lane 2 is not a contaminating band?

Minor points:

The authors should avoid using "degradation" in the model and discussion on PAT function in P2K1 regulation, as there is no experimental evidence for that. It is more accurate to use perhaps "turnover" or "protein level control" instead.

The last sentence of the discussion part does not fit in well with the rest of the section. "Hence, much remains to be discovered...environmental change." Consider some modification to this ending.

Methods and materials: Bacterial growth assays:

Line 426-428- "...the seedlings without roots or leaves were ground..."- change to "...either the leaves or the whole seedling without the root were ground..."

Reviewer #3 (Remarks to the Author):

In this study the authors uncover a mechanism by which Arabidopsis PRRs are negatively regulated through protein acylation. The authors identified via phosphoproteomics PAT5 and PAT9 as candidate substrates for the eATP receptor P2K1. Mutant plants lacking PAT5 and PAT9 exhibited enhanced calcium flux and ROS in responses to eATP, flg22, and chitin, and enhanced resistance to Pst bacteria. They further show that PAT5/9 interact with P2K1 at plasma membrane. In vitro phosphorylation assay showed that P2K1KD can phosphorylate PAT5/9 and that two candidate phosphosites are required for PAT9-mediated inhibition of immune responses (calcium flux, ROS, and antibacterial immunity). They further show that P2K1 is acylated in protoplasts in a manner dependent on PAT5 and PAT9 and this required the catalytic residues of PAT5/9. Interestingly, the P2K1 acylation in seedlings is induced upon eATP treatment. This induced acylation appeared to be correlated with protein turnover and decreased autophosphorylation of P2K1 in a manner dependent on PAT5/9. Several acylation sites were identified among which Cys394 and Cys407 are required for the turnover and decrease of autophosphorylation. Complementation experiments indicated that Xys394/407 are required for the negative regulation of eATP response and antibacterial immunity. The authors propose a model in which the PAT5/9 are activated by P2K1 upon stimulation by eATP. This leads to acylation of P2K1, which promotes turnover of the receptor protein and dampens signaling output. Overall, the work is novel and largely well done, and the subject is well-suited for Nature Communications. There are several issues, however, need to be addressed prior to publication.

Major comments:

1. The model proposes that the P2K1-mediated phosphorylation activates PAT5/9. There is no direct evidence this is the case. Yes, the authors showed that the phosphodead and phosphomimetic PAT9 variants modulate immune signaling positively and negatively, respectively. However, it remains unknown whether the P2K1 acylation requires phosphorylation on PAT5/9. It is necessary to show how the phosphodead and phosphomimetic PAT9 variants affect P2K1 acylation. These need to be done to support the model.
2. Are PAT5/9 required for the eATP-induced acylation? Fig 4a/b show constitutive acylation on P2K1, which is supposedly very very weak.
3. The characterization of PAT5/9 phosphosites is poorly done. The PAT5 and PAT9 phosphopeptides identified in phosphoproteomics (Fig1a) were completely different. S106 and S107 do not match the sequence of PAT9 (Fig 1a, Figs 3b, 3c, 3d). Are they referring to T107 and S109? Are these residues required for P2K1-mediated phosphorylation on PAT9?
4. It is not known whether the claimed phosphosites are conserved in PAT9 orthologs or paralogs in different species. This information is important as we need to know if similar mechanism may apply to species other than Arabidopsis. The authors showed that the C terminus of PAT5/9 is also

phosphorylated by P2K1 but do not discuss whether this is or is not involved in immune regulation.

5. Is PAT9 phosphorylation induced in vivo upon eATP stimulation?
6. Fig 4c. The right panel is confusing. The inhibition symbol is in a wrong place. I thought the inhibition is on P2K1, that is, reduced autophosphorylation and P2K1 protein level. Thus this symbol should be placed between the acyl-moiety and P2K1. The way it is drawn, it seems that the inhibitory action of P2K1 acylation leads to a strong defense.

Minor comments:

1. A phylogenetic tree of PATs should be provided. The data in Fig 1b/c/e seem to suggest redundant role of PATs, but it is not clear how similar are PAT5/9 and whether additional PATs may contribute to the induced acylation.
2. Fig 2b, the label of the 3rd lane from right is incorrect (should be a + for His-PAT9-c).
3. In Fig 3b/c/d, the S106/107A variant appeared to be partially functional. Please comment.
4. For bacterial growth assay, please box plot rather than box chart.
5. Title of Fig 3: "phosphorylate" should be "phosphorylates".

REVIEWER COMMENTS

Reviewer #1 (Remarks to the Author):

S-Acylation of P2K1 mediates extracellular ATP-induced immune signaling in Arabidopsis

Dongqin Chen^{1,2}, Fengsheng Hao¹, Nagib Ahsan^{3,4}, Jay J. Thelen³ and Gary Stacey^{2,3,5*}

Reviewer comments

This article studied the role of S-acylation in the regulation of phosphorylation and degradation of the P2K1 receptor in response of elicitor stimulation in Arabidopsis. The experiments were well designed and carried out. The data were well presented with proper statistic analysis. However, I have the following comments:

1. Interaction between PAT5/9 and P2K1. The authors showed clearly that interactions between PAT5/9-CRD, PAT5/9-C with P2K1 were independent to the autophosphorylation status of P2K1 (Fig 2). Then P2K1-KD strongly trans-phosphorylate PAT5/9-CRD, PAT5/9-C, but P2K1-KD1 did not. However, the two phosphosites of PAT9 are at S106 and S107, outside of the CRD and C-ter regions of PAT9. If the interactions between PAT5/9 and P2K1 are S-acylation and phosphorylation dependent can the authors explain how?

Answer: Thanks for asking this valuable question. The S106 and S107 are all in CRD domain of PAT9, we already showed them in the Supplementary Fig. 2d in the last version. We corrected them to “T107 and S109” in the manuscript due to the counting and writing mistake.

2. P2K1 S-acylation. S-acylation of P2K1 by PAT5/9 and the cys sites of P2K1 for S-acylation were confirmed by the Acyl-RAC assay using transfected Arabidopsis protoplasts (Fig 4). Can the authors indicate if the PM localization of P2K1 CS variants were also changed? In addition, because stable transgenic lines for PAT5/9:pat5/9, PAT5/9CS:pat5/9 and P2K1:p2k1 and P2K1CS:p2k1 were generated by the authors, I would recommend further confirmation for P2K1 S-acylation and localization using these stable transgenic lines

Answer: We appreciate the reviewer’s valuable suggestion. As shown in Supplementary Fig. 4c, the PM localization of P2K1 CS variants were the same as the wild-type P2K1 protein. Meanwhile, we crossed 35S:P2K1-GFP with *pat5* and *pat9* mutants, the P2K1 PM localization was not changed in either mutant background (Supplementary Fig. 4d). We also confirmed P2K1 S-acylation using stable transgenic plants (Fig. 4), but not NP:PAT9-C166S/*pat9* plant which was without any tag. The dynamic S-acylation levels of P2K1 in *pat5*, *pat9* mutants or phosphor-dead and phosphomimetic PAT9 variants background were also detected to support that PAT9 was required for the eATP-induced P2K1 acylation (Fig. 4d, e and Supplementary Fig. 4b).

3. Second T-DNA line for PAT5 and PAT9 should also be used

Answer: We appreciate the reviewer’s valuable suggestion. We confirmed the immune response using a second *Atpat9-2* mutant (SALK_206051C) (Fig. 1d and Supplementary Fig. 1e, h). For *Atpat5* T-DNA mutants, only the T-DNA insertion of GABI-322D08 was in the coding area, while GABI_827F02 was wrong and SALK_017032 was in the promoter (not affecting *PAT5* expression).

4. Line 200, 'directly phosphorylates and stimulates PAT5 and PAT9'.

You have only included the data for PAT9. Please also include data from PAT5

Answer: Thanks for this comment. We showed that P2K1 directly phosphorylated the PAT5-CRD or C domains in the kinase assay (Fig. 3a), but not the N-terminal of PAT5. The phosphosites of PAT5 identified in the KiC assay (Fig. 1a) were not conserved in the clade of PAT5, PAT6, PAT7, PAT8 and PAT9. However, the phosphosites of PAT9 identified by the KiC assay were in the CRD domain which was crucial for PAT9 S-acylation function, and were conserved in PAT9 orthologs or paralogs (Supplementary Fig. 2e). Therefore, we focused on exploring the role of PAT9 phosphorylation during PTI in *planta*. Although we understand the reviewer's curiosity about PAT5, including this additional data would not appreciably affect the impact of the paper.

5. Localization of PAT5 and PAT9 have been published by Batisctic, 2016, therefore, it is not necessary to include your data. However, it would be good to include your localization data from your stable transgenic lines as this has not been done before.

Answer: Thanks for this suggestion. Consistent with previous observation (Batisctic, 2016), PAT5 and PAT9 proteins are localized on the plasma membrane in the *PAT5-GFP* and *PAT9-GFP* stable transgenic plants (Supplementary Fig. 1b).

Minor comments:

Various spelling mistakes, misuse of scientific terms, bad construction of a few sentences, missing references and sources of reagents were found throughout the m/s. The authors need to carefully edit the manuscript before re-submitting.

1. Methods. Please refer to relevant references for individual methods followed, such as the acyl-RAC, pAM-PAT-35SS:YFP:GW, pGWB13 etc etc. Sources of flg22, chitin, sources of various antibodies, etc

Answer: As suggested, we added the relevant references and sources in the "Methods" section.

2. Line 180: please indicate conc and stained time as FM4-64 is NOT a PM specific dye and it only stains PM within the first few minutes before internalized

Answer: Thanks. We indicate the FM4-64 staining in "BiFC assay" of "Method" section.

Line 76: FLS2, use full term

Answer: Thanks. We revised it to "FLAGELLIN SENSING 2" which is the same as Perraki et al., 2018.

Line 80: 'lysin', change to lysine

Answer: The "lysin" is related to the CERK1 protein motif, it is not "lysine".

Line 81: 'CERK1', full term please

Answer: Thanks, we move the full term from line 82 to 81.

Line 82: 'The first plant, due to cell/tissue damage', badly constructed sentence, please change

Answer: Thanks. We revised.

Line 120: 'Of the 24 encoded AtPATs, only sevenplay roles in....', please add 'early seedling growth and establishment (AtPAT15)'

Answer: Thanks. We added.

Line 123: '....specific substrates for these AtPATs have not been identified ...'. This statement is not correct because CBL2,3 were identified as AtPAT10 substrates, for example (Zhou et al 2013).

Answer: Thanks, we revised. However, the CBL2,3 were shown to be AtPAT10 "putative substrates",

but not “specific or direct substrates”, as there were no interaction evidence presented, either in vitro or in vivo. Here is their conclusion: “Taken together, these results suggested that CBL2/3/6 are putative substrates of PAT10” (Zhou et al., 2013).

Line 467&478:(PH 7.5)...., change to (pH 7.5)

Answer: Thanks. We revised.

Line 480: at 20,000 xg....., change to 20,000 xg (g should be italic)

Answer: Thanks. We revised.

Line 494: .. (*p<0.05 or **p<0.01....., p should be italic)

Answer: Thanks. We revised.

Many more.....

Reviewer #2 (Remarks to the Author):

The manuscript titled “S-Acylation 1 of P2K1 mediates extracellular ATP-induced immune signaling in Arabidopsis” by Chen et al. is a very interesting study on the role of PAT5 and PAT9 (protein S-acyl transferases) in the regulation of immune responses mediated by the eATP receptor P2K1 (and possibly other PRRs as well). The authors found that the receptor P2K1 is able to phosphorylate PAT5 and PAT9 and this phosphorylation is rapidly lost upon ligand detection by P2K1. PAT5 and PAT9 are shown to be able to s-acylate P2K1. This P2K1 acylation mediated by PAT5 and PAT9 is required for the dephosphorylation of P2K1 and its return to its inactive steady state. This study shows a novel understanding about the regulation of PRR mediated plant immune responses by protein S-acyl transferases. The manuscript is well written, and experiments have been logically designed to answer key questions. There are some revisions to be made to the manuscript that will help to convey its message clearly.

Major points:

1. The authors should include a discussion of the interesting redundancy of the PAT5/9 clade. A simple phylogenetic tree should be included. The weak phenotypes of the pat5/9 single mutants may be explained by the redundancy of the family. Also, heterodimerization is observed in the single mutant, providing an explanation that even single mutants would exhibit observable phenotypes. Of course, to confirm these, the authors need to create the pat5/6/7, pat8/9, and pat5/6/7/8/9 combinatory mutants. The prediction would be that the 5/6/7 and 8/9 would show the same phenotype as 5/6/7/8/9 to support the heterodimerization model. As making these combinatory mutants takes a long time, I would agree it is out of the scope of the current study.

Answer: We appreciate reviewer’s understanding. We added a related discussion in our manuscript (Discussion section). Yes, the pat5/6/7/8/9 mutant will take a long time, so we only include a simple phylogenetic tree (Supplementary Fig. 2a). The phylogenetic relationship of PAT5/6/7/8/9 proteins present a rather high conservation rate. Meanwhile, we find that the phosphosites of PAT9 at T107 and S109 are highly conserved in PAT6 and PAT8 in *Arabidopsis*, or orthologous genes in rice, soybean and maize (Supplementary Fig. 2e).

2. In addition to the phylogeny, the authors should examine the existing microarray or RNA-seq datasets to examine the expression of the PAT5/9 clade. Are all 5 genes induced upon PTI/pathogen induction? Such analysis may help corroborate the whole story. The authors should also add the PAT dimerization in the model figure.

Answer: As suggested, we examined the relative gene expression of *PAT5*, *PAT6*, *PAT7*, *PAT8* and *PAT9* upon ATP or *Pst.* DC3000 treatment (Supplementary Fig. 2b). Our results showed that eATP or pathogen can induce *PAT5* to *PAT9* clade gene expression, suggesting they may be functionally redundant at the transcript level as at the protein level. Because the PAT proteins have 4 transmembrane domains and it is hard to draw them in the same PM site of our model figure, we prefer to revise "PAT" to "PAT dimer".

3. Figure 1

Panel d- include the double mutant *pat5 pat9* in the assay

Panel f- make it clear that the 15-minute sample alone was treated with Lambda PP to show that the higher band seen is the phosphorylated form of the protein. The CBB control bands are not the same in the samples representing different times after ATP treatment. The experiment should be repeated such that the loading is consistent in the lanes.

Panel g- in addition to the mock treated sample without Lambda PP treatment there must also be a control with the addition of the phosphatase

Legend: the *PAT9* complementation line has been incorrectly denoted as NP::ATPAT5-HA/Atpat5

Answer: Thanks for these valuable suggestions. As suggested, we included the MAPKs of *pat5/9* double mutant in Panel d, redid the *PAT5*-HA phosphorylation for Panel f, added the mock treated sample without Lambda PP treatment in the Panel g, and corrected the legend.

4. Supplementary Figure 2A

Panel A

In lines 175-176 of the manuscript it is stated that "Interestingly, all these interactions were reduced 15 min after elicitation and then recovered 1 hour after elicitation (Supplementary Fig. 2a)." In order to support this statement the 0 minute or prior to elicitation images of the *Nicotiana benthamiana* leaves need to be shown. Also, the figure should be organised better to make it easier to follow- the leaves showing the *PAT5* and *PAT9* interaction with P2K1 at different time points after elicitation (0, 15, 60 min post elicitation) could be placed together and the leaves showing other data such as *PAT5* and *PAT9*'s interaction with RBOHD and self and cross dimerization of the two PATs could be separate.

Answer: Thanks for this comment. The interactions between P2K1 and *PAT5/9* were reduced and then recovered in 1 hour upon eATP treatment in the LCI assay, but eATP can induce P2K1 protein degradation in 1 hour in *Arabidopsis* plant. Therefore, to avoid this mismatch and misunderstand, we decided to exclude this statement.

Lines 180-181: Interestingly, *PAT5* and *PAT9* appeared to form heterodimers and homodimers in both assays (Fig. 2a and Supplementary Fig. 2a)- There is no experiment shown in Figure 2A (BiFC) to show the formation of homodimers.

Answer: We appreciate reviewer's valuable suggestion. As suggested, we added the interactions between *PAT5* and *PAT5*, *PAT9* and *PAT9* which display their homodimers in the Figure 2A (BiFC).

5. Figure 3

Panel A- the Coomassie stained blot is not very clean and there are multiple contaminating bands of sizes similar to those of the proteins under study which makes discerning the right bands difficult. If possible, an anti-His blot can be performed which would give a much cleaner image of the input bands. Also, in the autoradiograph, in lanes 1 and 3 there is a band of size similar to that of His-PAT5-C. How do you know the band labelled as phosphorylated His-PAT5-C in lane 2 is not a contaminating band?

Answer: Thanks for asking this valuable question. First, the proteins of His-PAT5-CRD/C and His-PAT9-CRD/C contained non-specific bands in SDS page when purified from *E. coli*. We already checked their correct bands using anti-His blot (Fig. 2b). However, the concentration of $MnCl_2$ and $MgCl_2$ affects this kinase assay, which we carried out several times and this figure shows our best data.

Minor points:

The authors should avoid using “degradation” in the model and discussion on PAT function in P2K1 regulation, as there is no experimental evidence for that. It is more accurate to use perhaps “turnover” or “protein level control” instead.

Answer: Thanks. We revised to “turnover” in the model and discussion.

The last sentence of the discussion part does not fit in well with the rest of the section. “Hence, much remains to be discovered...environmental change.” Consider some modification to this ending.

Answer: Thanks. We revised.

Methods and materials: Bacterial growth assays:

Line 426-428- “...the seedlings without roots or leaves were ground...”- change to “...either the leaves or the whole seedling without the root were ground...”

Answer: Thanks. We revised it.

Reviewer #3 (Remarks to the Author):

In this study the authors uncover a mechanism by which Arabidopsis PRRs are negatively regulated through protein acylation. The authors identified via phosphoproteomics PAT5 and PAT9 as candidate substrates for the eATP receptor P2K1. Mutant plants lacking PAT5 and PAT9 exhibited enhanced calcium flux and ROS in responses to eATP, flg22, and chitin, and enhanced resistance to Pst bacteria. They further show that PAT5/9 interact with P2K1 at plasma membrane. In vitro phosphorylation assay showed that P2K1KD can phosphorylate PAT5/9 and that two candidate phosphosites are required for PAT9-mediated inhibition of immune responses (calcium flux, ROS, and antibacterial immunity). They further show that P2K1 is acylated in protoplasts in a manner dependent on PAT5 and PAT9 and this required the catalytic residues of PAT5/9. Interestingly, the P2K1 acylation in seedlings is induced upon eATP treatment. This induced acylation appeared to be correlated with protein turnover and decreased autophosphorylation of P2K1 in a manner dependent on PAT5/9. Several acylation sites were

identified among which Cys394 and Cys407 are required for the turnover and decrease of autophosphorylation. Complementation experiments indicated that Xys394/407 are required for the negative regulation of eATP response and antibacterial immunity. The authors propose a model in which the PAT5/9 are activated by P2K1 upon stimulation by eATP. This leads to acylation of P2K1, which promotes turnover of the receptor protein and dampens signaling output. Overall, the work is novel and largely well done, and the subject is well-suited for Nature Communications. There are several issues, however, that need to be addressed prior to publication.

Major comments:

1. The model proposes that the P2K1-mediated phosphorylation activates PAT5/9. There is no direct evidence this is the case. Yes, the authors showed that the phosphodead and phosphomimetic PAT9 variants modulate immune signaling positively and negatively, respectively. However, it remains unknown whether the P2K1 acylation requires phosphorylation on PAT5/9. It is necessary to show how the phosphodead and phosphomimetic PAT9 variants affect P2K1 acylation. These need to be done to support the model.

Answer: We appreciate the reviewer's valuable suggestion. As suggested, we crossed *PAT9-HA* plant with the phosphodead and phosphomimetic PAT9 variants plants, and then investigated the dynamic S-acylation levels of P2K1 upon eATP treatment (Fig. 4e). Indeed, phosphorylation of PAT9 affected ATP-elicited P2K1 receptor S-acylation (Fig. 4e).

2. Are PAT5/9 required for the eATP-induced acylation? Fig 4a/b show constitutive acylation on P2K1, which is supposedly very very weak.

Answer: Thanks for asking this valuable question. To support that PAT5/9 are required for the eATP-induced P2K1 S-acylation, we compared the dynamic S-acylation levels of P2K1 in the wild type with that in the *pat5* or *pat9* mutant backgrounds (Fig. 4d). Meanwhile, the dynamics of P2K1 S-acylation in the phosphodead and phosphomimetic PAT9 variants plants upon eATP treatment also support this conclusion (Fig. 4e).

3. The characterization of PAT5/9 phosphosites is poorly done. The PAT5 and PAT9 phosphopeptides identified in phosphoproteomics (Fig1a) were completely different. S106 and S107 do not match the sequence of PAT9 (Fig 1a, Figs 3b, 3c, 3d). Are they referring to T107 and S109? Are these residues required for P2K1-mediated phosphorylation on PAT9?

Answer: We appreciate that reviewer pointing out this issue. By checking the PAT9 protein sequence again, we find that the "S106 and S107" should be "T107 and S109", so we revise them in our manuscript. These two phosphosites of PAT9 were involved in stimulating the ATP-elicited Ca²⁺ influx and ROS production (Fig. 3b, c), mediating ATP-induced bacterial defense (Fig 3d), and PAT9 can be directly phosphorylated by P2K1 (Fig. 1a and Fig. 3a), demonstrating that P2K1 can directly phosphorylate and activate PAT5 and PAT9 to regulate ATP-induced PTI response through these two phosphosites. Meanwhile, the phosphodead and phosphomimetic PAT9 variants of these two phosphosites affected eATP-elicited P2K1 acylation (Fig. 4e). However, to maintain physiological levels of expression and translation of PAT9, the stable transgenic plants of phosphodead and phosphomimetic PAT9 variants were under the native promoter and without any tag, making it technically impossible to detect PAT9 phosphorylation upon ATP addition.

4. It is not known whether the claimed phosphosites are conserved in PAT9 orthologs or paralogs in different species. This information is important as we need to know if similar mechanism may apply to species other than *Arabidopsis*. The authors showed that the C terminus of PAT5/9 is also phosphorylated by P2K1 but do not discuss whether this is or is not involved in immune regulation.

Answer: Thanks for asking this valuable question. Yes, the phosphosites are conserved in PAT9 orthologs or paralogs in different species. We compared PAT9 with PAT5, PAT6, PAT7 and PAT8, also with Glyma.10g222900, Glyma.02g079900 (soybean), LOC_Os08g42370, LOC_Os08g42620 (rice), GRMZM2G016805 and GRMZM2G100641 (maize), most of these four sites are conserved in *Arabidopsis* or other species (Supplementary Fig. 2e).

5. Is PAT9 phosphorylation induced in vivo upon eATP stimulation?

Answer: Yes, please see PAT9-HA phosphorylation in Fig. 1g, addition of Lambda protein phosphatase could release PAT9-HA phosphate groups.

6. Fig 4c. The right panel is confusing. The inhibition symbol is in a wrong place. I thought the inhibition is on P2K1, that is, reduced autophosphorylation and P2K1 protein level. Thus this symbol should be placed between the acyl-moiety and P2K1. The way it is drawn, it seems that the inhibitory action of P2K1 acylation leads to a strong defense.

Answer: Thanks for this comment. Our Fig. 4c is about the dynamic P2K1 S-acylation upon ATP treatment, it doesn't contain a right panel. We showed that S-acylation of P2K1-HA was remarkably increased after elicitation (Fig. 4c top panel), which only included S-acylated P2K1 protein but not the total P2K1 autophosphorylation and protein level. This, together with our other additional data (Figs. 4c, 5 and Supplementary Figs. 4, 5), indicates that the S-acylation of P2K1 receptor has an inverse relationship to P2K1 autophosphorylation and turnover, acting as a negative regulator of innate immunity which might insure a steady-state inactive state for P2K1 that insures against spurious activation.

Minor comments:

1. A phylogenetic tree of PATs should be provided. The data in Fig 1b/c/e seem to suggest redundant role of PATs, but it is not clear how similar are PAT5/9 and whether additional PATs may contribute to the induced acylation.

Answer: Thanks. Because phylogeny of *Arabidopsis* PAT proteins was showed by Batistic (2012), we added a simple phylogenetic tree in *Arabidopsis* (Supplementary Fig. 2a) and PAT9 phosphosites in rice, soybean and maize (Supplementary Fig. 2e).

2. Fig 2b, the label of the 3rd lane from right is incorrect (should be a + for His-PAT9-c).

Answer: Thanks. We revised.

3. In Fig 3b/c/d, the S106/107A variant appeared to be partially functional. Please comment.

Answer: Thanks. We comment in the "Result" section.

4. For bacterial growth assay, please box plot rather than box chart.

Answer: Thanks, we changed Figure 1e to a box chart. For Figure 3d and 6b, we would like to compare "Mock" with "ATP" in the same mutant, and then between different mutant plants, so we prefer to put "Mock" with "ATP" chart side by side using a box chart.

5. Title of Fig 3: "phosphorylate" should be "phosphorylates".

Answer: Thanks. We revised.

REVIEWERS' COMMENTS

Reviewer #1 (Remarks to the Author):

I'm happy with the revised m/s. Thanks for taking your time to carefully address all the points raised.

Reviewer #2 (Remarks to the Author):

The authors have addressed almost all my concerns. From my point of view, it is ready for publishing. Congratulations on the ground-breaking and thorough study!

Reviewer #3 (Remarks to the Author):

The authors have done an outstanding job revising the manuscript. I am particularly happy to see new data provided in Fig. 4d and 4e, which are critical for the main conclusion. Altogether the study provides important new insight in the regulation of receptor dynamics.

REVIEWERS' COMMENTS

Reviewer #1 (Remarks to the Author):

I'm happy with the revised m/s. Thanks for taking your time to carefully address all the points raised.

Answer: We appreciate the reviewer's comment.

Reviewer #2 (Remarks to the Author):

The authors have addressed almost all my concerns. From my point of view, it is ready for publishing. Congratulations on the ground-breaking and thorough study!

Answer: We appreciate the reviewer's comment.

Reviewer #3 (Remarks to the Author):

The authors have done an outstanding job revising the manuscript. I am particularly happy to see new data provided in Fig. 4d and 4e, which are critical for the main conclusion. Altogether the study provides important new insight in the regulation of receptor dynamics.

Answer: We appreciate the reviewer's comment.